# Intelligent Diagnosis of Rolling Bearings Fault Based on Multisignal Fusion and MTF-ResNet

**DOI:** 10.3390/s23146281

**Published:** 2023-07-10

**Authors:** Kecheng He, Yanwei Xu, Yun Wang, Junhua Wang, Tancheng Xie

**Affiliations:** 1School of Mechatronics Engineering, Henan University of Science and Technology, Luoyang 471003, China; 13323859953@163.com (Y.W.); wangjh@haust.edu.cn (J.W.); xietc@haust.edu.cn (T.X.); 2Intelligent Numerical Control Equipment Engineering Laboratory of Henan Province, Luoyang 471003, China

**Keywords:** metro traction motor bearings, multisignal fusion, Markov transition field, optimized deep residual network, diagnosis of compound faults

## Abstract

Existing diagnosis methods for bearing faults often neglect the temporal correlation of signals, resulting in easy loss of crucial information. Moreover, these methods struggle to adapt to complex working conditions for bearing fault feature extraction. To address these issues, this paper proposes an intelligent diagnosis method for compound faults in metro traction motor bearings. This method combines multisignal fusion, Markov transition field (MTF), and an optimized deep residual network (ResNet) to enhance the accuracy and effectiveness of diagnosis in the presence of complex working conditions. At the outset, the acquired vibration and acoustic emission signals are encoded into two-dimensional color feature images with temporal relevance by Markov transition field. Subsequently, the image features are extracted and fused into a set of comprehensive feature images with the aid of the image fusion framework based on a convolutional neural network (IFCNN). Afterwards, samples representing different fault types are presented as inputs to the optimized ResNet model during the training phase. Through this process, the model’s ability to achieve intelligent diagnosis of compound faults in variable working conditions is realized. The results of the experimental analysis verify that the proposed method can effectively extract comprehensive fault features while working in complex conditions, enhancing the efficiency of the detection process and achieving a high accuracy rate for the diagnosis of compound faults.

## 1. Introduction

As the power source of metro trains, the quality of the traction motor bearings directly affects the normal operation of the motor. The frequent starting and stopping of the metro causes alternating changes in the speed of the traction motor bearings and the loads they are subjected to. With long-term harsh working conditions, the inner and outer rings of bearings and rolling elements will produce varying degrees of pitting, cracking and more complex forms of failure. The adverse vibrations generated by a faulty bearing, when input into the entire system over an extended period, not only damage the traction motor but also pose a risk to other structural components. This poses a serious threat to the safety and reliability of metro trains. The intelligent diagnosis of bearings fault in complex working conditions enables the timely identification of fault types, facilitating early maintenance intervention and providing significant engineering value for practical applications.

Conventional approaches for bearing fault diagnosis predominantly rely on signal processing techniques. To address the issue of noise interference during feature extraction, wavelet thresholding was employed to effectively eliminate significant noise components from the raw data [1,2]. In an effort to enhance the signal-to-noise ratio, ref. [3,4] adopted empirical mode decomposition (EMD) to decompose the signal into multiple intrinsic mode functions. Furthermore, ref. [5] introduced an optimized variational mode decomposition (VMD) method to facilitate the selection of intrinsic mode functions containing pertinent fault information. Despite the promising outcomes achieved by these traditional methods in bearing fault diagnosis, they are accompanied by inherent limitations. These drawbacks encompass restricted generalization capability, challenges in extracting deep fault features, and complexities associated with parameter optimization. Signal analysis technology, as a research hotspot, has been receiving attention from scholars. Subsequently, the introduction of new methods has successfully addressed many challenges [6,7].

With the development of artificial intelligence technology, machine learning and deep learning [8] have gained significant attention in various fields, and numerous researchers have started extracting deeper features and making notable contributions [9,10,11]. A convolutional neural network (CNN), as one of their important representatives, possesses a powerful adaptive feature extraction capability. Moreover, CNN has demonstrated remarkable performance in the field of image processing. As such, scholars have increasingly introduced CNN into the field of fault diagnosis and conducted a series of research studies in this area. Ref. [12] has recently proposed a CNN model that utilizes widened convolutional kernels to improve the feature extraction efficiency of the network. Ref. [13] has deployed a CNN to extract features from Mel spectrum generated from the voiceprint signals of motors. Ref. [14] has presented a multiscale CNN model that effectively extracts signal features at different frequencies. This advanced model is further combined with LSTM to identify fault types. In the field of medical imaging, ref. [15] proposed an improved CNN model architecture for the identification of a lung nodule and early-stage cancer diagnosis by comparing multiple photos. In big data environments, to reduce the costs associated with data collection and processing, some researchers have explored unsupervised learning techniques. To synchronously extract local and global structural information from the raw unlabeled industrial data, ref. [16] proposed a new multiple-order graphical deep extreme learning machine (MGDELM) algorithm. Ref. [17] proposed a novel self-training semi-supervised deep learning (SSDL) approach to train a fault diagnosis model together with few labeled and abundant unlabeled samples. The previously discussed research studies have made notable advances in fault diagnosis. However, because of their reliance on single-sensor signals, there may be limitations in accurately characterizing fault information, which could ultimately reduce their overall reliability.

Multisignal fusion technology enables the simultaneous processing of time-series data obtained from multiple sensors, thereby capturing a broader range of system variability while offering heightened complementarity and fault tolerance. In one study, feature extraction was performed on original vibration and acoustic signals, which were subsequently fused using a 1DCNN-based network model [18]. Another approach proposed a frequency-domain multilinear principal component analysis to effectively identify faults by integrating diverse vibration and acoustic signals [19]. Similarly, a two-dimensional matrix was constructed from multi-axial vibration signals, and an enhanced 2DCNN model was employed for fault diagnosis [20]. These methods have demonstrated commendable enhancements in diagnostic accuracy. However, it is worth noting that a limitation common to these approaches is the omission of time correlation among signals, which may result in the loss of crucial fault-related information.

Upon a comprehensive analysis of existing literature, it has been observed that diagnostic approaches leveraging deep learning techniques frequently employ increasing network depths to enhance the model’s learning capacity and improve diagnostic performance. Nevertheless, the utilization of progressively deeper networks may give rise to challenges such as the vanishing or exploding gradient problem. To address this issue, deep residual networks were introduced [21], effectively mitigating the aforementioned problem. Furthermore, an innovative activation function named STAC-tanh was proposed by [22], which enables adaptive feature extraction in the bearing system by employing the hyperbolic tangent function with slope and threshold adaptivity. Another compelling approach involved the fusion of Gramian angular field (GAF) with ResNet, leading to notable advancements in bearing fault diagnosis [23]. Additionally, ref. [24] combined transfer learning with ResNet, utilizing a pretrained ResNet model on ImageNet as a fault feature extractor, which yielded remarkably accurate results. These aforementioned studies have demonstrated promising outcomes in the realm of bearing fault diagnosis. However, certain limitations persist, including the sole reliance on a single sensor signal and the absence of experimental verification through the use of a purpose-built platform.

In summary, most of the studies are based on open source datasets with simple working conditions and failure forms, but the actual working conditions of bearings are complex and can present different parts and degrees of failure. To address the challenges faced in compound bearing fault diagnosis under complex working conditions, such as the low reliability of single sensor signals, the tendency for traditional data processing methods to result in important information loss, the degradation of diagnostic models with increasing network depth, and the difficulty of feature extraction, this paper proposes an intelligent diagnosis method for compound bearing faults in metro traction motors by combining MTF-processed acoustic-vibration signals using IFCNN for feature fusion along with an optimized version of ResNet. The main contributions of the paper are expressed as follows:The application of IFCNN in compound bearing fault diagnosis allows for the fusion of multiple signal features, reducing the limitations of single sensor signals and providing more reliable diagnostic results.The optimized ResNet model improves the efficiency of feature extraction by addressing the vanishing gradient problem. Combined with the MTF data processing method, it can effectively extract complex bearing fault features under varying working conditions with good accuracy and stability.The construction of a test platform for metro traction motor bearings was completed, and intelligent diagnosis of composite faults under variable working conditions was conducted, validating the effectiveness of the proposed methods.

The remaining sections of this paper are arranged as follows: In Section 2, the data processing method used in this study and the construction of the dataset are introduced. Section 3 focuses on the multisignal fusion technology used in this study. Section 4 provides a detailed description of the fault diagnosis model and the corresponding diagnostic process. Section 5 explains the specific experimental design, as well as the diagnostic scheme adopted in this study. Section 6 analyzes the experimental results and carries out a series of method comparisons to validate the effectiveness of the proposed approach. Section 7 summarizes the main content of the paper and draws conclusions.

## 2. Data Preprocessing

In this study, a signal acquisition system will be built to obtain a large amount of raw data using acoustic emission sensors, vibration sensors and PCI acquisition cards. The research focuses on compound faults, with pitting as the main defect. The location of the defect is used as a classification criterion. A total of eight fault types including normal bearings are designed and labeled for subsequent study, using different fault locations as classification indicators. The fault types and labels are shown in Table 1.

### 2.1. Dataset Construction

The vibration and acoustic emission signals were acquired using a PCI data acquisition card with a sampling frequency of 50 kS/s and a sampling time of 10 s, giving a total of 5 × 10^5^ sampling points. In this experiment, the minimum speed of the bearing is determined to be 800 rpm. Based on this speed, the number of sampling points obtained from one cycle of bearing rotation can be calculated to be 3750. In order to ensure the completeness of the sampled fault information, it is recommended that the number of sampling points be at least twice that of the calculated value, resulting in a sampling length of 8192 (2^13^). With a limited amount of data, the vibration and acoustic emission signals were data augmented using overlapping sampling so that each fault type under each working condition contained 1000 samples for a total of 8000 samples, which were randomly divided into a training set and a testing set at 9:1. Under fixed working conditions, the dataset is divided as shown in Table 2.

### 2.2. MTF Image Encoding

In this paper, MTF is used to process vibration signals and acoustic emission signal data, converting the acquired data samples into image samples. MTF is an image encoding method that converts original vibration or acoustic emission signals into time series two-dimensional images through Markov transition probabilities [25].

Suppose a discretized segment of time series data $X=\left\{x_{1},x_{2},\cdots ,x_{n}\right\}$ is partitioned into intervals of its value domain by quantile Q. Each $x_{t}$ in the sequence can be mapped to the corresponding interval $q_{n}(n\in [1,Q])$. By calculating the state transfer probabilities through the Markov chain principle, a state transfer probability matrix W of size Q×Q can be obtained, with an expression, as shown in Equation (1), where $w_{ij}$ denotes the probability that a sample point in interval $q_{j}$ at moment t is transferred to interval $q_{i}$ at moment t+1 [26].
(1)$$W=\left[\begin{array}{ccc} w_{11\left|P\left(x_{t+1}\in q_{1}\left|x_{t}\in q_{1}\right.\right)\right.} & \cdots & w_{1Q\left|P\left(x_{t+1}\in q_{1}\left|x_{t}\in q_{Q}\right.\right)\right.}\\ w_{21\left|P\left(x_{t+1}\in q_{2}\left|x_{t}\in q_{1}\right.\right)\right.} & \cdots & w_{2Q\left|P\left(x_{t+1}\in q_{2}\left|x_{t}\in q_{Q}\right.\right)\right.}\\ \vdots & \ddots & \vdots \\ w_{Q1\left|P\left(x_{t+1}\in q_{Q}\left|x_{t}\in q_{1}\right.\right)\right.} & \cdots & w_{QQ\left|P\left(x_{t+1}\in q_{Q}\left|x_{t}\in q_{Q}\right.\right)\right.} \end{array}\right]$$

By incorporating the temporal information into the state transfer probability matrix W and arranging each state transition probability $w_{ij}$ in time sequence, a Markov transition field (MTF) matrix M of size n×n is obtained as expressed in shown Equation (2) where $m_{ij}$ denotes the transition probability $w_{ij}$ between the intervals $\left(q_{j}\to q_{i}\right)$ in which the sample points are located in time sequence.
(2)$$M=\left[\begin{array}{cccc} m_{11} & m_{12} & \cdots & m_{1n}\\ m_{21} & m_{22} & \cdots & m_{2n}\\ \vdots & \vdots & \ddots & \vdots \\ m_{n1} & m_{n2} & \cdots & m_{nn} \end{array}\right]=\left[\begin{array}{ccc} w_{ij\left|x_{1}\in q_{i},x_{1}\in q_{j}\right.} & \cdots & w_{ij\left|x_{1}\in q_{i},x_{n}\in q_{j}\right.}\\ w_{ij\left|x_{2}\in q_{i},x_{1}\in q_{j}\right.} & \cdots & w_{ij\left|x_{2}\in q_{i},x_{n}\in q_{j}\right.}\\ \vdots & \ddots & \vdots \\ w_{ij\left|x_{n}\in q_{i},x_{1}\in q_{j}\right.} & \cdots & w_{ij\left|x_{n}\in q_{i},x_{n}\in q_{j}\right.} \end{array}\right]$$

The elements $m_{ij}$ in the MTF matrix are transformed as pixel points into a two-dimensional feature image with temporal correlation. As the number of sample points selected directly affects the size of the generated coded image, it is clearly inappropriate for an image with too large a size to be used directly as input to the CNN. To improve computational efficiency, a fuzzy kernel ${\left\{\frac{1}{m^{2}}\right\}}_{m\times m}$ is used to pixel average each region without overlap. Figure 1 shows images of different fault types after encoding each sample, consisting of 8192 sampling points, using MTF image encoding and subsequently subjecting them to pixel averaging processing. Compared to traditional time domain analysis methods, MTF encoding images preserve time-related information and enable clearer differentiation of various fault types in rolling bearings.

## 3. Multisignal Fusion

To enhance system stability and increase diagnostic reliability, this article collected vibration signals and acoustic emission signals and fused them for processing. This fusion processing can establish correlations between multiple signal sources. Usually, information fusion can be divided into three levels: data-level fusion, feature-level fusion, and decision-level fusion. Considering that the sample data in this study consist of MTF encoded images of different fault types, it is advantageous to employ CNN for image processing. Therefore, this paper adopted the IFCNN for feature-level fusion of the data.

IFCNN consists of three modules, namely, the feature extraction module, the feature fusion module and the feature reconstruction module [27], and the structure of this framework is shown in Figure 2.

The feature extraction module consists of two convolutional layers. The first layer uses the first convolutional layer of the ResNet101 network model, pretrained on the ImageNet dataset. This layer includes 64 convolutional kernels with a size of 7 × 7 and retains the training parameters, enabling effective extraction of image features. The second convolutional layer includes 64 convolutional kernels with a size of 3 × 3, which are used to adjust the features extracted by the first layer in order to adapt to feature fusion. For this study, the feature fusion module adopts an element-wise maximum fusion strategy. The final module is the image reconstruction module, in which the third convolutional layer includes 64 convolutional kernels with a size of 3 × 3. This layer adjusts the fused convolutional features and plays an important role in reconstructing the image. The fourth convolutional layer reconstructs the feature map with three-channel output, and it includes 3 convolutional kernels with a size of 1 × 1.

This framework uses the mean squared error (MSE) as the basic loss function and adds a perceptual loss to optimize the model. The expression for the perceptual loss ($P_{loss}$) is as follows:(3)$$P_{loss}=\frac{1}{C_{f}H_{f}W_{f}}{\sum_{i,x,y}\left[f_{p}^{i}\left(x,y\right)-f_{g}^{i}\left(x,y\right)\right]}^{2}$$
where $f_{p}$ and $f_{g}$ are the feature maps of the predicted fused image and the true fused image, respectively; i is the feature map channel index; $C_{f}$, $H_{f}$ and $W_{f}$ are the number of channels, height and width of the feature map, respectively. The expression for the basic loss ($B_{loss}$) is as follows:(4)$$B_{loss}=\frac{1}{3H_{g}W_{g}}{\sum_{i,x,y}\left[I_{p}^{i}\left(x,y\right)-I_{g}^{i}\left(x,y\right)\right]}^{2}$$
where $I_{p}$ and $I_{g}$ are the predicted fused image and the true fused image, respectively; i is the RGB image channel index; $H_{g}$ and $W_{g}$ are the height and width of the true fused image, respectively. The expression for the total loss ($T_{loss}$) is as follows:(5)$$T_{loss}=w_{1}B_{loss}+w_{2}P_{loss}$$
where $w_{1}$ and $w_{2}$ are the weighting coefficients. For the fusion of MTF-encoded images in this study, the sums are both set to 1.

## 4. Fault Diagnosis Method

### 4.1. Optimized Deep Residual Network

ResNet is built on the basis of CNN and solves the gradient vanishing problem by adding skip connections between the input and output of each convolutional layer. The classic residual module structure is shown in Figure 3.

The structure contains two mappings, the part of the main path is called the residual mapping and the part of the bypass connection is called the constant mapping. The final output of the residual block is therefore the superposition of the outputs obtained from the two mappings:(6)H(x)=F(x)+x

The structure of the residual network model constructed in this study is shown in Table 3. It includes an input layer, a maximum pooling layer, convolutional layers, an average pooling layer, a fully connected layer and a softmax classifier. Conv2, Conv3, Conv4 and Conv5 are residual modules.

Convolutional layers are the core of CNNs, responsible for extracting features from large amounts of input data. Typically, convolutional layers can be described by the following expression:(7)$$x_{j}^{l}=\sigma (\sum_{i\in M_{j}}x_{j}^{l-1}*k_{ij}^{l}+b_{j}^{l})$$
where $x_{j}^{l-1}$ is the input of the (l−1)-th layer of the network; $x_{j}^{l}$ is the output of the l-th layer of the network; $k_{ij}^{l}$ is the weight matrix of the convolutional kernel; $b_{j}^{l}$ is the bias term; $M_{j}$ is the set of input feature maps; σ is the nonlinear activation function; and ∗ represents the convolution operation.

Pooling aims to reduce the size of feature maps while retaining the most important feature information. It can effectively reduce computational complexity and improve the model’s robustness and generalization capabilities. The pooling process involves four steps: input feature map, sliding window coverage, feature aggregation, and output feature map. The pooling process can be described by the following expression:(8)$$x_{j}^{l}=\sigma (\beta_{j}^{l}down(x_{j}^{l-1})+b_{j}^{l})$$
where $x_{j}^{l-1}$ is the input of the (l−1)-th layer of the network; $x_{j}^{l}$ is the output of the l-th layer of the network; $b_{j}^{l}$ is the bias term; σ is the nonlinear activation function; down(⋅) is the down-sampling function; and $\beta_{j}^{l}$ is the weight.

To improve the efficiency of fault diagnosis, a convolutional block attention module (CBAM) is introduced to optimize the model by focusing it more on important features [28]. CBAM consists of channel attention module, which captures the connections between channels of the feature map, and spatial attention module, which captures the connections between spatial regions of the feature map.

The channel attention module feeds the features $F_{avg}^{c}$ and $F_{\max}^{c}$ obtained after using average pooling and max pooling in the channel dimension into the convolutional network, respectively, and sums the results and outputs them. The process is described as:(9)$$M_{c}(F)=\sigma (W_{1}(W_{0}(F_{avg}^{c}))+W_{1}(W_{0}(F_{\max}^{c})))$$
where σ is a sigmoid function; $W_{0}$ and $W_{1}$ are convolution operations with a convolution kernel size of 1 × 1.

The spatial attention module performs a convolution operation on the features $F_{avg}^{s}$ and $F_{\max}^{s}$ obtained after stitching using average pooling and max pooling in the channel dimension. The process is described as:(10)$$M_{s}(F)=\sigma (f^{7\times 7}([F_{avg}^{s};F_{\max}^{s}]))$$
where σ is a sigmoid function; $f^{7\times 7}$ is convolution operation with a convolution kernel size of 7 × 7.

This study introduced CBAM into ResNet without changing the overall structure of the network. The input data are MTF feature images of size 224 × 224. After passing through the first convolutional layer with a kernel size of 7 × 7 and a stride of 2, the image size is reduced to 112 × 112. This is followed by a max pooling layer with a stride of 2, which further reduces the data dimensionality and the image size to 56 × 56. The channel attention and spatial attention modules are added sequentially after the batch normalization (BN) layer at the end of the residual modules Conv2, Conv3, Conv4 and Conv5, respectively. After passing through the Conv2, which has 64 channels and convolutional kernels of size 3 × 3 with a stride of 1, deeper features are extracted while maintaining the same image size as the previous layer. The channels in Conv3, Conv4, and Conv5 are doubled successively to 128, 256 and 512. At the same time, down-sampling is implemented in the first convolutional layer with a stride of 2 in each residual module. This results in output image sizes that progressively decrease to 28 × 28, 14 × 14 and 7 × 7, respectively. Afterwards, the network passes through an average pooling layer to reduce the number of parameters and mitigate the occurrence of overfitting. Then, a fully connected layer is used for nonlinear combination of the extracted features, followed by a softmax classifier to produce the final output.

The proposed model uses a cross-entropy loss function to evaluate the error between the predicted and true values, avoiding gradient dispersion, which is defined in the context of a multiclassification problem as:(11)$$L=\frac{1}{N}\sum_{i}L_{i}=-\frac{1}{N}\sum_{i}\sum_{c=1}^{M}y_{ic}\log(p_{ic})$$
where M is the number of categories; $y_{ic}$ is the sign function, taking 1 if the true value of sample i is equal to c and 0 otherwise; and $p_{ic}$ is the predicted probability that sample i belongs to category c.

An initial test was carried out with a constant speed of 1600 rpm and a load of 7 kN, the number of epochs was set to 50 and the loss and accuracy (Acc) in training are shown in Figure 4.

Overall, from the graph, it can be seen that when the epoch reaches 40, the loss and accuracy have basically converged, and the accuracy has reached nearly 100%. This indicates that the model performs well on the training set and has good generalization ability, which also verifies that the model structure and parameters chosen in this paper are correct. Setting the number of epochs too large can significantly prolong the training time and even cause overfitting, while setting it too small may not find the global optimal solution. After multiple tests, this paper set the learning rate to 0.001 and the number of epochs to 40, which is a good choice. To intuitively demonstrate the advantages of the proposed method in extracting fault features, this paper utilized the uniform manifold approximation and projection (UMAP) algorithm to perform dimensionality reduction on the data and visualize the results. Taking the steady state condition with a speed of 1600 rpm and a load of 7 kN as an example, this paper conducted a layer-by-layer analysis of ResNet models with and without CBAM and extracted the output features of the intermediate layers for calculation. Then, UMAP is utilized to reduce the dimensionality of the extracted features to two dimensions. This paper extracted the fault features from the avgpool layer and visualized the results using a scatter plot where different fault types are marked with different colors. The visualization is shown in Figure 5.

As can be seen from the figure above, there is a significant difference in the clustering degree of data samples between the two models, and introducing CBAM to ResNet can yield more obvious clustering effect in the avgpool layer. Therefore, it can be concluded that the proposed optimized ResNet has excellent abilities in extracting fault features under complex working conditions.

### 4.2. Fault Diagnosis Process

This paper proposes a compound fault diagnosis method of rolling bearings based on multisignal fusion and MTF-ResNet. The fused MTF-encoded images are input into the ResNet model for training, and the fault is intelligently diagnosed under different working conditions. The basic process is shown in Figure 6, and the main steps are as follows: (1) acquire vibration and acoustic emission signals; (2) generate feature images of size 224 × 224 by MTF encoding of the original data to build a training set and a test set; (3) fuse the MTF encoded images of the two signals using IFCNN; (4) input the training set into the optimized ResNet model built for training, and save the optimal parameters; and (5) test the test samples and output the results to complete the intelligent fault diagnosis.

## 5. Fault Diagnosis Experiment

### 5.1. Experimental Design

The experimental bearing was selected as NU216 cylindrical roller bearing. Defects were artificially introduced to the inner and outer rings, as well as the rolling elements using a YLP-MDF-152 laser marking machine from Han’s Laser. Taking into account the failure mechanism of bearings in actual working environments, alternating loads can cause cracks to form at a certain depth below the surface, which may then propagate to the surface and cause spalling. Fatigue spalling increases vibration and noise during rotation and is usually the main form of rolling bearing failure. Therefore, pitting was produced on the surface of the bearing at different locations to simulate early defects. The pitting diameter was set to 40 μm and the depth was set to 30% of the laser energy. Eight types of faults, as described in Section 2, were designed using different fault positions as classification criterion.

In order to simulate the working conditions of metro traction motors, three additional speeds and three additional loads were included in the experimental design. In consideration of both actual working conditions and minimizing the impact of bearing degradation on the experiment, gradient speeds of 800 rpm (low), 1600 rpm (medium) and 2400 rpm (high) were chosen, along with gradient equivalent dynamic loads of 5 kN (light), 7 kN (medium) and 9 kN (heavy) as the radial loads. There are a total of 72 (8 × 3 × 3) subexperiments. The experimental arrangement is shown in Table 4.

### 5.2. Construction of the Signal Acquisition System

This study utilized the intelligent testing platform for comprehensive bearing performance, jointly developed by Henan University of Science and Technology, Luoyang Bearing Research Institute, and Intelligent Numerical Control Equipment Henan Provincial Engineering Laboratory, as the signal acquisition system. The testing machine allows for a maximum inner diameter of 120 mm, a maximum speed of 5000 r/min, a maximum radial load of 300 kN, and a maximum axial load of 200 kN for the bearing. The platform is equipped with a PCI-8 acoustic emission transmitter, two R50S-TC acoustic emission sensors, two LC0151T acceleration sensors, two LC0201-5 signal conditioners, and a PCI8510 data acquisition card.

During the experiment, a healthy bearing and a faulty bearing were installed at both ends of the testing machine’s spindle, and vibration and acoustic emission signals were collected from both bearings simultaneously. The loading system applies radial loads to the spindle via a pair of NU2218 cylindrical roller bearings, which in turn are transferred to the test bearings at both ends of the spindle. The sensor signals are amplified and conditioned by signal amplifiers, signal conditioners, and input to the computer through a PCI acquisition card. The principle of the signal acquisition system is shown in Figure 7. The physical set-up of the system is shown in Figure 8.

### 5.3. Diagnostic Scheme Design

To further validate the effectiveness of the proposed method, three types of diagnostic schemes were designed for single working condition changes, compound working condition changes, and generic working conditions, considering two different factors (speed and load) that affect the test results.

When studying single working condition changes, first control the speed to be constant, put data of two different loads in the training set, and put data of another load in the test set to verify the robustness of the model. When controlling the load to be constant, the method is similar to the above. The specific diagnostic program is shown in Table 5.

When studying the change of compound working condition, it is required that the training set contains data with different speeds and loads at the same time. For generic working conditions, it is required that all fault types data under all conditions exist in both the training and testing sets.

## 6. Experimental Results and Comparison of Methods

During the operational process of a metro system, variations in bearing speed and load are inevitable. While previous steady-state tests have certain limitations, it becomes crucial to analyze the results of variable working condition tests to validate the effectiveness of the proposed method. To further explore the changes in compound working conditions, an additional analysis comparing the fusion of acoustic emission and vibration signals with a single signal was incorporated to emphasize the advantages of the proposed method. In the generic working condition tests, the feature extraction capabilities of four models, namely the proposed model, RepVGG, CBAM-CNN and ResNet, were compared to evaluate their performance.

### 6.1. Single Working Condition Changes

Based on the fault diagnosis method proposed in Section 5.3, with the control of constant speed and load, the training set was input into the model constructed in this paper, and fault diagnosis was performed on the test set. The diagnostic results are shown in Table 6.

Based on a comprehensive examination of the aforementioned table, it is observed that when maintaining a constant speed while altering the load, the fault diagnosis accuracy reaches nearly 100%. Conversely, in cases where the load remains constant but the speed varies, a decrease in fault diagnosis accuracy is observed, indicating a substantial influence of rotational speed on diagnostic outcomes. Subsequent analysis reveals that the accuracy of items numbered 12, 15 and 18 is significantly low, whereas items numbered 3, 6 and 9 demonstrate accuracy close to 100%, albeit slightly lower than other items within the initial nine numbers. This discrepancy can be attributed to the fact that fault characteristics extracted under medium- to high-speed and medium to heavy load conditions are more discernible compared to those under low-speed and light load conditions.

### 6.2. Compound Working Condition Changes

Mixed data with different speeds and loads were included in the training set and used to train the model proposed for fault diagnosis on the testing set. Subsequently, a comparison was made between the fusion of acoustic emission and vibration signals and using a single signal. The diagnostic results are shown in Table 7.

The table clearly indicates that the diagnostic results of items numbered 4 to 6 surpass those of items numbered 1 to 3. Notably, the training and testing sets for items numbered 1 to 3 encompass varying rotation speeds, whereas items numbered 4 to 6 involve different loads. It is observed that the diagnostic accuracy of items numbered 4 to 6 remains relatively stable, whereas item numbered 3 exhibits significantly lower accuracy compared to items numbered 1 and 2. The underlying reason behind this phenomenon aligns with the findings presented in Section 6.1 of this paper.

From the standpoint of signal acquisition, the fusion of acoustic emission and vibration signals yields higher diagnostic accuracy in fault diagnosis compared to utilizing a single signal. This finding provides further substantiation that the application of multisignal fusion technology can effectively enhance system stability and diagnostic accuracy. Furthermore, it is evident that employing a single vibration signal for diagnostics yields superior results in comparison to employing a single acoustic emission signal. This can be attributed to the fact that the acoustic emission acquisition system exhibits heightened sensitivity to environmental noise, primarily stemming from the operational testing equipment, which poses challenges in noise elimination.

### 6.3. Generic Working Conditions

To evaluate the performance of the proposed fault diagnosis model, all fault samples involving three different speeds and three different loads were included in both the training and testing sets. The sample ratio between the two sets was set to 9:1 to ensure the training set was large enough to enable the model to effectively learn the fault data while still reserving an adequate number of samples for testing. Subsequently, the model was applied to diagnose faults on the testing set. To visualize the diagnostic results, a confusion matrix was employed, providing an intuitive and reliable representation of classifications made by the model. The confusion matrix is presented in Figure 9.

The confusion matrix provides a clear and intuitive visualization of the model’s misclassifications and the types of errors. It can be seen that the overall diagnostic performance is good, and the accuracy rate for the fusion of acoustic emission and vibration signals is almost 100%. However, the diagnosis accuracy rate for label 6, which corresponds to the “outer Ring + rolling element pitting” fault type, is relatively low. The model misclassified three test samples as “rolling element pitting”. Further analysis revealed that the two types of faults have similar features, making it difficult to extract differences between them. By comparing (a–c) in Figure 9, the results further confirm that multisignal fusion technology has higher reliability and accuracy compared to a single signal, especially under changing working conditions.

To compare the feature extraction capabilities of different models, the training and testing sets samples of above-mentioned generic working conditions were respectively input into RepVGG, CBAM-CNN and ResNet models for diagnosis. Two types of faults were selected as examples: label 1 (corresponding to “inner ring pitting”) with better diagnostic results and label 6 (corresponding to “outer ring + rolling element pitting”) with poorer results. The precision–recall (PR) curves and receiver operating characteristic (ROC) curves were generated for the optimized ResNet, RepVGG, CBAM-CNN and ResNet models and evaluation indicators, such as average precision (AP) and area under the curve (AUC) were introduced.

The precision–recall (PR) curve is a graphical representation of the performance of a binary classification model, with recall on the x-axis and precision on the y-axis. It illustrates the trade-off between precision and recall at various classification thresholds. The relevant theoretical formulas for the PR curve are as follows:(12)$$Precision=\frac{TP}{TP+FP}$$
(13)$$Recall=\frac{TP}{TP+FN}$$
where *TP* represents the number of true positive instances; *FP* represents the number of false positive instances; and *FN* represents the number of false negative instances.

The principle of average precision (AP) is to summarize the Precision-Recall (PR) curve by calculating the average precision value. It can be obtained by computing the area under the PR curve. It provides a comprehensive assessment of how well the model balances precision and recall across different recall levels.

The receiver operating characteristic (ROC) curve is a tool used to evaluate the performance of binary classification models. It plots the false positive rate (*FPR*) on the x-axis and the true positive rate (*TPR*) on the y-axis. The principle of the ROC curve can be described using the following formulas:(14)$$TPR=\frac{TP}{TP+FN}$$
(15)$$FPR=\frac{FP}{FP+TN}$$
where *FP* represents the number of negative instances incorrectly classified as positive; *TN* represents the number of negative instances correctly classified as negative; *TP* represents the number of positive instances correctly classified as positive; and *FN* represents the number of positive instances incorrectly classified as negative.

Area under the curve (AUC) is obtained by calculating the area under the ROC curve. The resulting AUC value ranges from 0 to 1, where 0.5 represents a random classifier and 1 represents a perfect classifier. A higher AUC value indicates better classifier performance.

The diagnostic results are presented in the form of PR and ROC curves in Figure 10 and Figure 11. The overall accuracy rate, AP and AUC for all fault types were calculated for the four models, and the weighted average values were recorded in Table 8.

Generally, the closer the PR curve in Figure 10 is to the upper right corner, the larger the AP value, and the better the model performance. The closer the ROC curve in Figure 11 is to the upper left corner, the larger the AUC value, and the better the model performance. Observing the figure above, it can be seen that for the two selected fault types with different diagnostic effects, the PR and ROC curves of proposed model are both closer to the right-angle edge than those of RepVGG, CBAM-CNN and ResNet, indicating better performance. Combined with the data in Table 8, the three accuracy evaluation indicators of the proposed model are higher than those of the compared models, validating the good feature extraction ability of the proposed model.

## 7. Conclusions

This paper focused on the study of the feature extraction ability of the model for complex working conditions, using the metro traction motor bearings as the research object. On the basis of ResNet, CBAM was introduced to optimize the ResNet model. Nine different working conditions and eight compound fault types were designed for experimentation. In addition, a dataset was constructed using MTF image encoding and IFCNN image fusion technology. During the model training process, UMAP was used for visualization to intuitively demonstrate the feature extraction effect of the proposed model. After the experiment, three evaluation indicators were used for objective evaluation of the feature extraction ability of the optimized ResNet, RepVGG, CBAM-CNN and ResNet models.

The results of the experiment show that the MTF-ResNet model with multisignal fusion performs well under complex working conditions, with a diagnostic accuracy rate of up to 99.25%. Based on the results, some important conclusions can be drawn. Specifically, in terms of sensors, using only vibration signals produces better diagnostic results than using only acoustic emission signals. In addition, compared with a single signal, using acoustic emission and vibration signal fusion can provide more comprehensive and integrated information, while reducing misclassifications caused by the limitations of a single signal, thereby improving fault diagnosis accuracy and making the diagnosis result more reliable. In terms of data processing, MTF image encoding technology is a simple data processing method that retains the time correlation of the data, making it easier for the model to extract more comprehensive fault features. For feature extraction models, introducing CBAM after the batch normalization layers of the ResNet model can make the model more focused on capturing important features, quickly distinguishing different types of fault features, and improving diagnostic efficiency. Furthermore, the ResNet structure can effectively alleviate the gradient disappearance phenomenon that occurs as the network deepens, thereby preventing model degradation.

Undoubtedly, this study presents several avenues for future research in the proposed methodologies. Firstly, the inclusion of additional sensors or exploration of different sensor types holds promise. For instance, incorporating multidirectional vibration sensors or temperature sensors could offer a more comprehensive spectrum of fault information, thereby enhancing diagnostic fault tolerance. Secondly, exploring more advanced data processing techniques warrants investigation to enhance the quality of input signals. The acoustic emission signals acquired in this study exhibited significant levels of environmental noise that proved challenging to eliminate. Therefore, employing sophisticated techniques may substantially improve the value derived from these acoustic emission signals. Moreover, conducting model testing on larger datasets utilizing more complex compound faults can effectively confirm the feature extraction capabilities and generalization of the model. This approach will serve as a more robust means of validation. Furthermore, future research focusing on feature extraction models should prioritize the development of lightweight and efficient models to facilitate practical implementation.

Despite the inherent limitations of the methods proposed in this paper, they exhibit commendable feature extraction capabilities within intricate operational scenarios. Consequently, these methods hold potential for application in fault diagnosis tasks related to metro traction motor bearings, thereby possessing appreciable value in engineering applications.

## Figures and Tables

**Figure 1 sensors-23-06281-f001:**
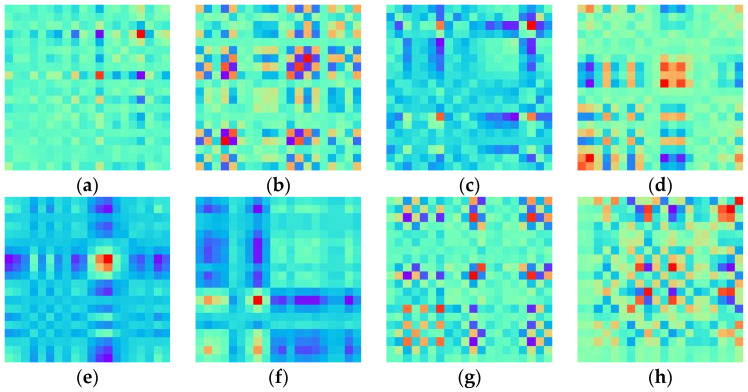
MTF-encoded images of 8 types of fault: (**a**) normal; (**b**) inner ring; (**c**) outer ring; (**d**) rolling element; (**e**) inner ring + outer ring; (**f**) inner ring + rolling element; (**g**) outer ring + rolling element; (**h**) inner ring + outer ring + rolling element.

**Figure 2 sensors-23-06281-f002:**
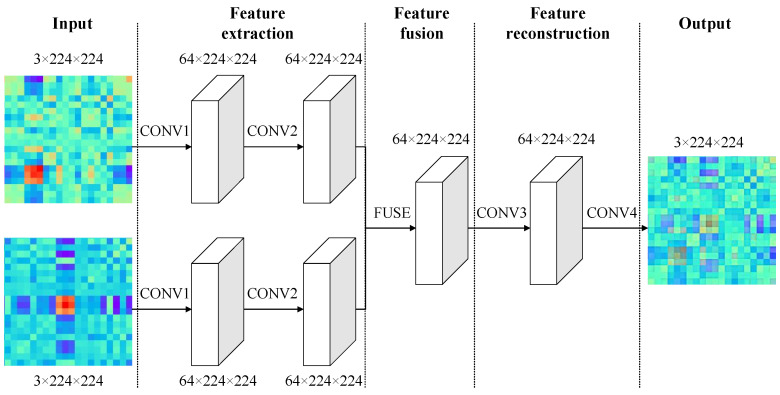
The structure of IFCNN.

**Figure 3 sensors-23-06281-f003:**
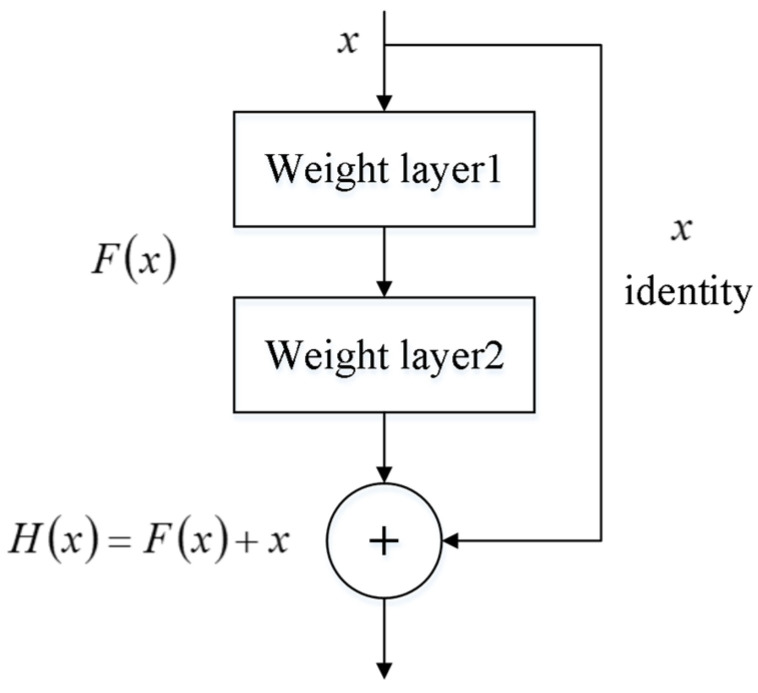
Residual module structure.

**Figure 4 sensors-23-06281-f004:**
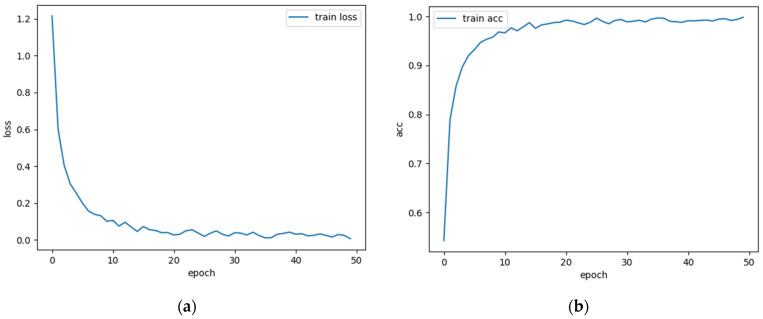
Loss and accuracy in training with 50 epochs: (**a**) loss; (**b**) acc.

**Figure 5 sensors-23-06281-f005:**
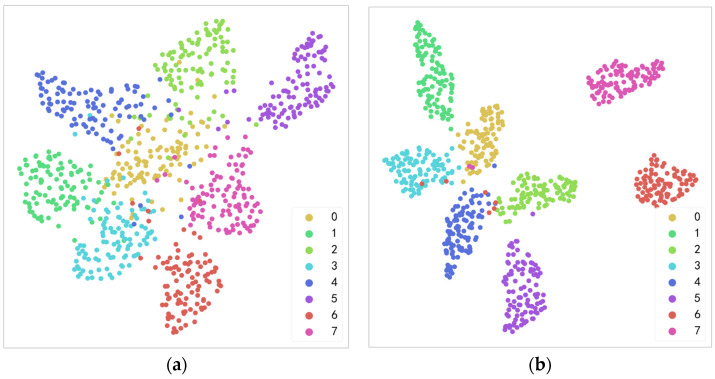
Visualization of fault features extracted from the avgpool layer: (**a**) before the introduction of CBAM; (**b**) after the introduction of CBAM.

**Figure 6 sensors-23-06281-f006:**
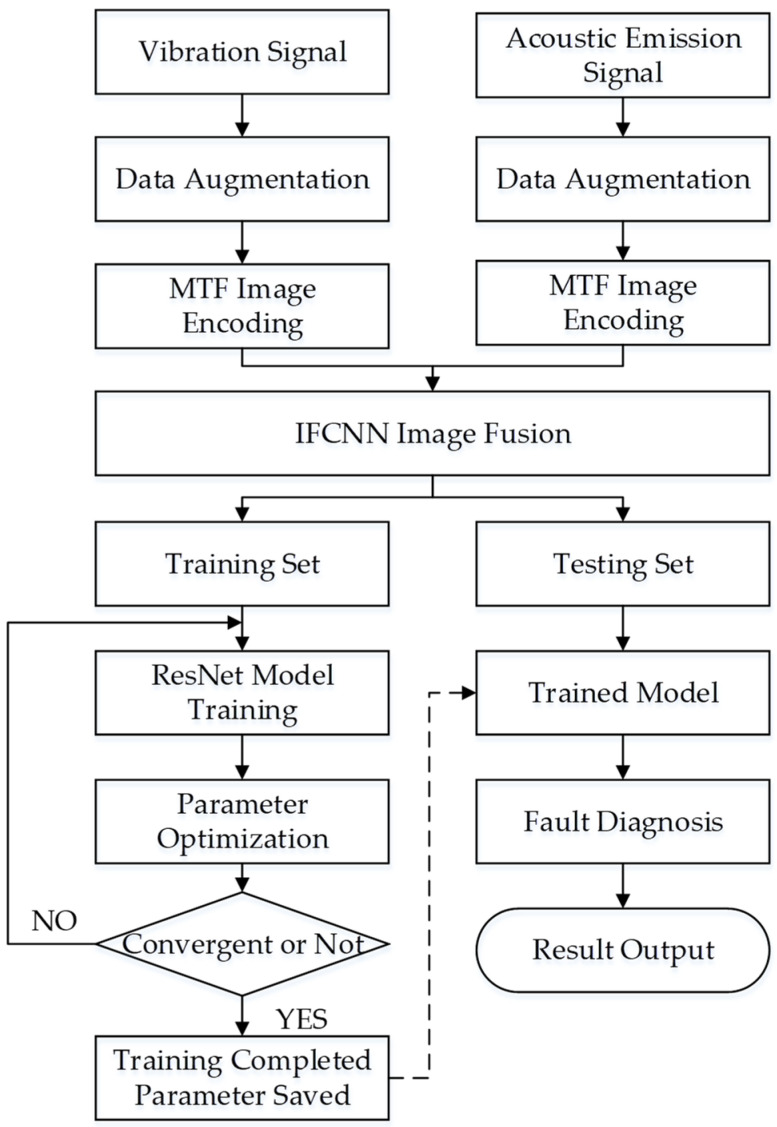
Fault diagnosis process of rolling bearings based on multisignal fusion and MTF-ResNet.

**Figure 7 sensors-23-06281-f007:**
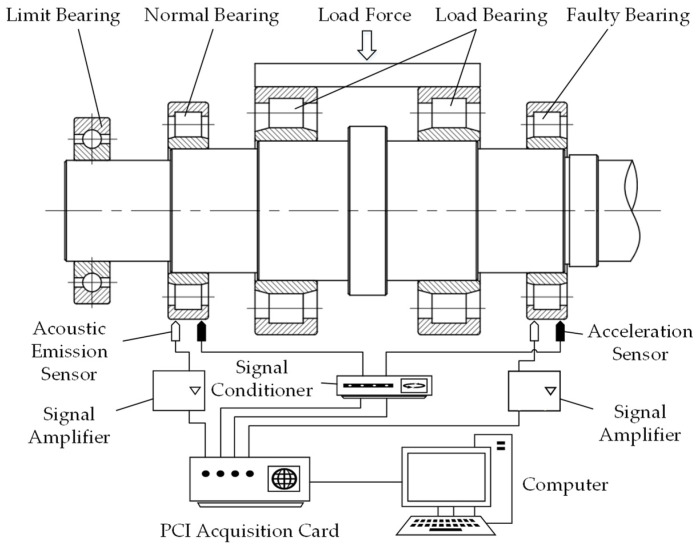
Schematic diagram of the signal acquisition system.

**Figure 8 sensors-23-06281-f008:**
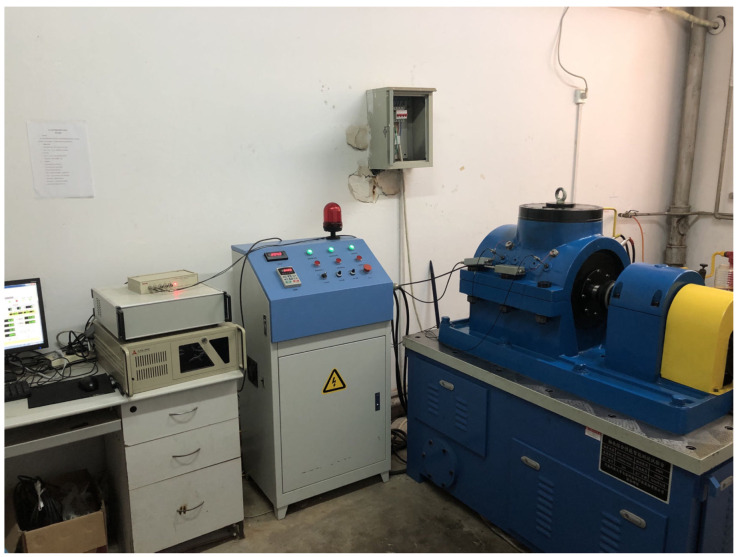
Photograph of the built signal acquisition system.

**Figure 9 sensors-23-06281-f009:**
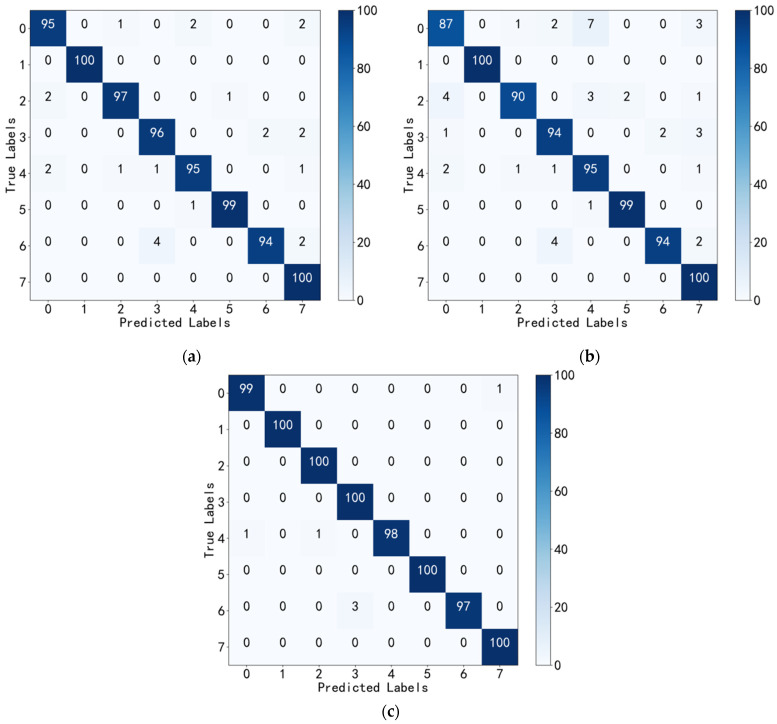
The diagnostic results for generic working condition: (**a**) based on vibration signal (with an accuracy rate of 97%); (**b**) based on acoustic emission signal (with an accuracy rate of 94.88%); (**c**) based on the fusion of acoustic emission and vibration signals (with an accuracy rate of 99.25%).

**Figure 10 sensors-23-06281-f010:**
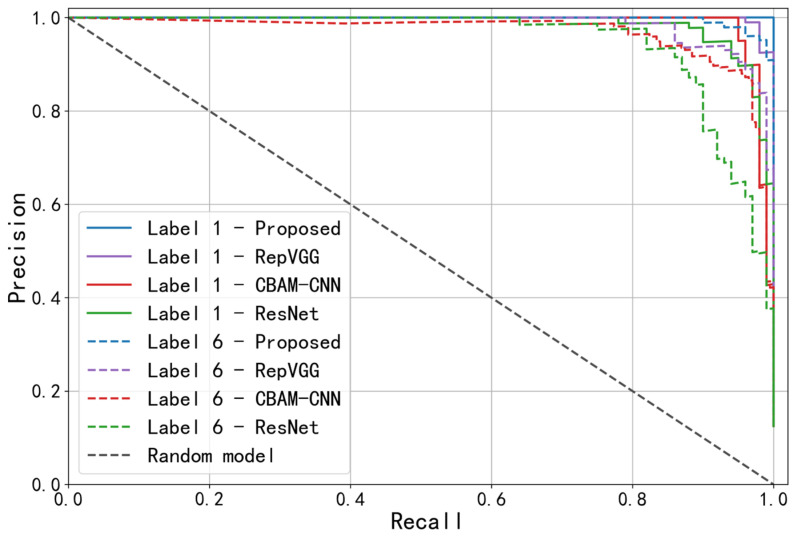
PR curves of four models with two fault types.

**Figure 11 sensors-23-06281-f011:**
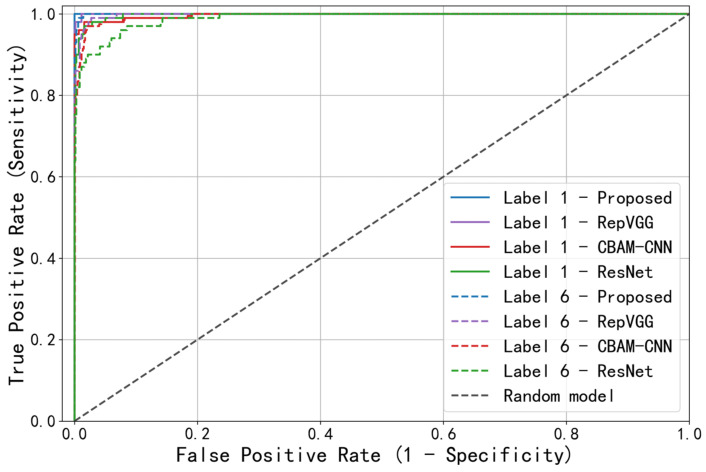
ROC curves of four models with two fault types.

**Table 1 sensors-23-06281-t001:** Label settings for different fault types.

	Fault Types							
	Normal	Inner Ring	Outer Ring	Rolling Element	Inner Ring + Outer Ring	Inner Ring + Rolling Element	Outer Ring + Rolling Element	Inner Ring + Outer Ring + Rolling Element
Label	0	1	2	3	4	5	6	7

**Table 2 sensors-23-06281-t002:** Dataset partitioning under fixed working conditions.

Data Set	Sample Size for Different Fault Types							
	Normal	Inner Ring	Outer Ring	Rolling Element	Inner Ring + Outer Ring	Inner Ring + Rolling Element	Outer Ring + Rolling Element	Inner Ring + Outer Ring + Rolling Element
Training Set	900	900	900	900	900	900	900	900
Testing Set	100	100	100	100	100	100	100	100

**Table 3 sensors-23-06281-t003:** ResNet model structure.

Layer Name		Kernel Size	Channel	Stride	Padding	Output
Input Image		-	-	-	-	3 × 224 × 224
Conv1		7 × 7	64	2	3	64 × 112 × 112
Maxpool		3 × 3	64	2	1	64 × 56 × 56
Conv2	Conv2_1	3 × 3	64	1	1	64 × 56 × 56
	Conv2_2	3 × 3	64	1	1	64 × 56 × 56
	Conv2_3	3 × 3	64	1	1	64 × 56 × 56
	Conv2_4	3 × 3	64	1	1	64 × 56 × 56
Conv3	Conv3_1	3 × 3	128	2	1	128 × 28 × 28
	Conv3_2	3 × 3	128	1	1	128 × 28 × 28
	Conv3_3	3 × 3	128	1	1	128 × 28 × 28
	Conv3_4	3 × 3	128	1	1	128 × 28 × 28
Conv4	Conv4_1	3 × 3	256	2	1	256 × 14 × 14
	Conv4_2	3 × 3	256	1	1	256 × 14 × 14
	Conv4_3	3 × 3	256	1	1	256 × 14 × 14
	Conv4_4	3 × 3	256	1	1	256 × 14 × 14
Conv5	Conv5_1	3 × 3	512	2	1	512 × 7 × 7
	Conv5_2	3 × 3	512	1	1	512 × 7 × 7
	Conv5_3	3 × 3	512	1	1	512 × 7 × 7
	Conv5_4	3 × 3	512	1	1	512 × 7 × 7
Avgpool		-	-	-	-	512 × 1 × 1
Fc, Softmax						

**Table 4 sensors-23-06281-t004:** Experimental arrangement.

Speed/rpm	Radial Loads for Different Fault Types/kN							
	Normal	Inner Ring	Outer Ring	Rolling Element	Inner Ring + Outer Ring	Inner Ring + Rolling Element	Outer Ring + Rolling Element	Inner Ring + Outer Ring + Rolling Element
800	5	5	5	5	5	5	5	5
800	7	7	7	7	7	7	7	7
800	9	9	9	9	9	9	9	9
1600	5	5	5	5	5	5	5	5
1600	7	7	7	7	7	7	7	7
1600	9	9	9	9	9	9	9	9
2400	5	5	5	5	5	5	5	5
2400	7	7	7	7	7	7	7	7
2400	9	9	9	9	9	9	9	9

**Table 5 sensors-23-06281-t005:** Diagnostic scheme for single working condition change.

Fixed Variable	Variable	Training Set	Testing Set
Speed/rpm	Load/kN	5, 7	9
		5, 9	7
		7, 9	5
Load/kN	Speed/rpm	800, 1600	2400
		800, 2400	1600
		1600, 2400	800

**Table 6 sensors-23-06281-t006:** The diagnostic results for single working condition changes.

No.	Speed of Training Set/rpm	Load of Training Set/kN	Speed of Testing Set/rpm	Load of Testing Set/kN	Diagnostic Accuracy/%
1	800	5, 7	800	9	99.5
2	800	5, 9	800	7	100
3	800	7, 9	800	5	93
4	1600	5, 7	1600	9	100
5	1600	5, 9	1600	7	99.4
6	1600	7, 9	1600	5	92.4
7	2400	5, 7	2400	9	100
8	2400	5, 9	2400	7	100
9	2400	7, 9	2400	5	94
10	800, 1600	5	2400	5	100
11	800, 2400	5	1600	5	100
12	1600, 2400	5	800	5	78
13	800, 1600	7	2400	7	98.6
14	800, 2400	7	1600	7	90.4
15	1600, 2400	7	800	7	81
16	800, 1600	9	2400	9	98.8
17	800, 2400	9	1600	9	99.4
18	1600, 2400	9	800	9	83

**Table 7 sensors-23-06281-t007:** The diagnostic results for compound working condition changes.

No.	Speed of Training Set/rpm	Load of Training Set/kN	Speed of Testing Set/rpm	Load of Testing Set/kN	Diagnostic Accuracy/%		
					Vibration	Acoustic Emission	Fusion
1	800, 1600	5, 7, 9	2400	5, 7, 9	90	82	94.1
2	800, 2400	5, 7, 9	1600	5, 7, 9	93.4	88.1	97.6
3	1600, 2400	5, 7, 9	800	5, 7, 9	71.4	66	75
4	800, 1600, 2400	5, 7	800, 1600, 2400	9	98	90.5	100
5	800, 1600, 2400	5, 9	800, 1600, 2400	7	96.5	85	99.4
6	800, 1600, 2400	7, 9	800, 1600, 2400	5	97.1	83.4	98.6

**Table 8 sensors-23-06281-t008:** The accuracy evaluation indicators of the four models.

Model	Evaluation Indicator		
	Accuracy/%	AP	AUC
Proposed	99.25	0.989	1.000
RepVGG	96.72	0.967	0.996
CBAM-CNN	94.16	0.953	0.993
ResNet	88.35	0.935	0.988

## Data Availability

The data presented in this study are available on request from the author.

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
