# Peer review of "Intelligent Diagnosis of Rolling Bearings Fault Based on Multisignal Fusion and MTF-ResNet"

_sensors, 2023, doi:10.3390/s23146281_

Round 1

Reviewer 1 Report

This paper proposes an intelligent diagnosis method for compound faults in metro traction motor bearings based on multi-signal fusion and optimized deep residual network. It is of interest. However, the authors should address the following points to further improve the quality.

1. The contributions of this paper are not clear. They should be highlighted at the end of the introduction. In addition, more information on the background and problem statement should be added.

2. The structure of the paper is not clear. For example, “Experimental Platform and Methodology” only introduce the experimental platform. Moreover, it is best to introduce this part in the experimental section.

3. Check the place of “3.1. Design of Diagnostic Programme”

4. The performance of the proposed approach should be compared with more state-of-the-art fault detection approaches.

5. Literature review on the deep learning-based methods of fault diagnosis is limited. More recently-published papers in this field should be discussed. It is recommended to add some such as Multiple-Order Graphical Deep Extreme Learning Machine (MGDELM), self-training semi-supervised deep learning (SSDL) and other latest reference and algorithms to strengthen the understanding of different deep learning-based intelligent fault diagnosis methods.

6. The linguistic quality needs improvement. I can find some typos.

The linguistic quality needs improvement. I can find some typos.

Reviewer 2 Report

The manuscript titled "Intelligent Diagnosis of Rolling Bearings Fault Based on Multi-2 Signal Fusion and MTF-ResNet" fits the aims and scopes of the journal. iT looks an original contribution to the literature for Sensors. Scientific contribution is sufficient. Although there are many positives about the manuscript but yet I feel that the manuscript needs some minor revision before its publication. 

1. The authors should avoid to use too much abbreviations in the abstract.

2. Introduction needs improvement in terms of literature review and motivation for the presented work.

3. Language quality needs some refinement.

4. The authors should not use bulk number of references in one go e.g. [1-5], instead each reference should be discussed separately.

5. Authors should take care of copyright issues regarding use of figures in the manuscript.

6. Equation (8) in line 251 needs explanation in detail.

7. Conclusion section should briefly contain future scope of studies.

Minor refinement is required.

Reviewer 3 Report

The author has made an effort to improve his paper, but many issues still need to be addressed.

There are some points given below that need to be addressed

1-      There are many too many long sentences in every section, e.g., at the start of the abstract, the first sentence is almost five lines, which is not a good sign for the reader and the researcher. Please try to cut these sentences into small meaningful chunks

2-      At the start of the introduction, the author used two different styles of references, e.g., name and numeric value, while according to the MDPI format author needs to use only numeric value. I suggest the author strictly follow the MDPI format

3-      At the end of the introduction, part flow of the study is missing. How does a reader know what is discussed in the sections

4-      In the manuscript was, many observations used. I suggest the author add a table at the end of the introduction in which observation and its full form are added.

5-      In line number 129,185,343,362,377 table numbers and figure numbers are missing .please correct all these.

6-      Follow the MDPI stretcher strictly and rearrange the paper as I saw a result section is added, but there is no number and heading for the result, and I am a little bit confused to see the paper arrangement.

7-      If you take figures from somewhere, add references, as I saw some figures already present in different papers.

8-      Different equations are present in the paper, but some overlap because of improper use of math tools.

I suggest you write equations in math.h software where you find everything according to math

9-      A literature review portion is missing in the paper, and I suggest adding this section and studying some papers related to your work. Some related papers are given below.

Mazhar, T., Irfan, H. M., Khan, S., Haq, I., Ullah, I., Iqbal, M., & Hamam, H. (2023). Analysis of Cyber Security Attacks and Its Solutions for the Smart Grid Using Machine Learning and Blockchain Methods. Future Internet15(2), 83.

Haq, I., Mazhar, T., Malik, M. A., Kamal, M. M., Ullah, I., Kim, T., ... & Hamam, H. (2022). Lung Nodules Localization and Report Analysis from Computerized Tomography (CT) Scan Using a Novel Machine Learning Approach. Applied Sciences12(24), 12614.

Rahmani, A. M., Ali, S., Yousefpoor, M. S., Yousefpoor, E., Naqvi, R. A., Siddique, K., & Hosseinzadeh, M. (2021). An area coverage scheme based on fuzzy logic and shuffled frog-leaping algorithm (sfla) in heterogeneous wireless sensor networks. Mathematics9(18), 2251.

Naqvi, R. A., Hussain, D., & Loh, W. K. (2021). Artificial Intelligence-Based Semantic Segmentation of Ocular Regions for Biometrics and Healthcare Applications. Computers, Materials & Continua66(1).

A lot og grammer mistake found in paper. A complete review of english expert is needed to improve paper .

Reviewer 4 Report

A comparative literature review was not conducted. The exact purpose of the study is not stated and why it was found is not well explained in detail.

The paper's english is poor. 

Round 2

Reviewer 1 Report

The revision has addressed all my issues. The quality has improved a lot after revision. I recommend it for publication.

The quality of English language is fine

Author Response

Dear Reviewer,

Thank you for your diligent review and valuable suggestions. Your suggestions have greatly helped improve our research capabilities. We appreciate your agreement to publish our article in your esteemed journal. We will continue to work hard and strive for more innovative achievements.

Kind regards,

Kecheng He

Reviewer 3 Report

Please carefully read all comments and try to solve all. 

The author has tried to improve his paper and solved comments, but some issues still need to be addressed.

1-      At the end of the introduction, the author added a flow of study in which the author used a word chapter. Please confirm that the authors are writing a thesis or research paper. Change word chapter to Section.

2-      I also advised the author to add a literature review portion after the introduction, which is still missing. Please carefully address all comments and follow the reviewer's instructions.

3-      Still, equations need to be readable; suggest using math.h tool to draw the equation

4-      In the manuscript was, many observations used. I suggest the author add a table at the end of the introduction in which observation and its full form are added. This is still missing  

Minor English proof reading is needed.

Reviewer 4 Report

No more comments

it is better than before

Author Response

(The authors gave the same response as above.)
